# An Evolving TinyML Compression Algorithm for IoT Environments Based on Data Eccentricity

**DOI:** 10.3390/s21124153

**Published:** 2021-06-17

**Authors:** Gabriel Signoretti, Marianne Silva, Pedro Andrade, Ivanovitch Silva, Emiliano Sisinni, Paolo Ferrari

**Affiliations:** 1UFRN-PPgEEC, Postgraduate Program in Electrical and Computer Engineering, Federal University of Rio Grande do Norte, Natal 59078-970, Brazil; gabrielsig@ufrn.edu.br (G.S.); pedro.meira.055@ufrn.edu.br (P.A.); 2UNIBS-DIE, Department of Information Engineering, University of Brescia, 25123 Brescia, Italy; emiliano.sisinni@unibs.it (E.S.); paolo.ferrari@unibs.it (P.F.)

**Keywords:** internet of things, online data compression, TinyML, eccentricity, evolving algorithm, LPWAN

## Abstract

Currently, the applications of the Internet of Things (IoT) generate a large amount of sensor data at a very high pace, making it a challenge to collect and store the data. This scenario brings about the need for effective data compression algorithms to make the data manageable among tiny and battery-powered devices and, more importantly, shareable across the network. Additionally, considering that, very often, wireless communications (e.g., low-power wide-area networks) are adopted to connect field devices, user payload compression can also provide benefits derived from better spectrum usage, which in turn can result in advantages for high-density application scenarios. As a result of this increase in the number of connected devices, a new concept has emerged, called TinyML. It enables the use of machine learning on tiny, computationally restrained devices. This allows intelligent devices to analyze and interpret data locally and in real time. Therefore, this work presents a new data compression solution (algorithm) for the IoT that leverages the TinyML perspective. The new approach is called the Tiny Anomaly Compressor (TAC) and is based on data eccentricity. TAC does not require previously established mathematical models or any assumptions about the underlying data distribution. In order to test the effectiveness of the proposed solution and validate it, a comparative analysis was performed on two real-world datasets with two other algorithms from the literature (namely Swing Door Trending (SDT) and the Discrete Cosine Transform (DCT)). It was found that the TAC algorithm showed promising results, achieving a maximum compression rate of 98.33%. Additionally, it also surpassed the two other models regarding the compression error and peak signal-to-noise ratio in all cases.

## 1. Introduction

The Internet of Things (IoT) allows the continuous monitoring of environments and machines using small and inexpensive sensors [1,2]. From this perspective, advances in the technologies of sensors, microcontrollers, and communication protocols have made the mass production of IoT platforms with many connectivity options possible, and at affordable prices [3,4,5]. Due to the low cost of IoT hardware, the sensors are being deployed on a large scale to monitor machines and, in general, public and residential spaces [6,7,8].

These sensors monitor the physical properties associated with their deployment environments—twenty-four hours a day, seven days a week—making them a critical facilitator to increase the volume of the data generated, due to the collection performed in a wide variety of scenarios [9,10]. Note that these data streams can form a large volume of data very rapidly, and as a consequence, they generate the need for a vast amount of disk storage space [11,12]. For example, accelerometers embedded in vehicles are constantly collecting vibration patterns for various types of roads. Likewise, several other sensors such as magnetometer, pressure, temperature, humidity, and ambient light sensors, among many others, measure the physical conditions wherever they are installed, in addition to those that have been developed for the health area [8,13].

Therefore, it can be noted that the generation of data by the sensors occurs in a continuous stream format, which has varying characteristics according to each application. The sample rate, for example, may vary from once a day to every millisecond depending on the context [14]. This variation in the collection occurs because there are several distinct application demands, different device models, and different configuration needs or limitations in the data transmission channel [15].

In this scenario, it is essential to study and use techniques that can compress the data, as well as algorithms and functionalities that can be embedded in these devices. In the last few decades, much research in the specific area of data compression in IoT-based monitoring systems has been published [9,11,16], focusing on different layers and proposing different strategies. Generally, compression aims to eliminate redundant information in order to minimize the space required for storage, consequently reducing the number of bits to be transmitted periodically to the endpoint, in addition to improving efficiency in the information analysis [9,17].

In this manner, some challenges regarding the amount of data generated per second are encountered, especially when using low-power transmission protocols as LoRaWAN [18,19,20]. Furthermore, it is known that most of the battery consumption of a device that communicates using a wireless medium is in the transmission of the data and not in the processing of the information, which makes it advantageous in terms of energy to compress data locally before transmission [16,21,22].

Often, IoT devices work with low-frequency processors and have limited local data storage capacity, severe memory restrictions [23,24], simple protocols, and inefficient medium access strategies (e.g., pure ALOHA). Additionally, when operating in unlicensed bands, the bandwidth is scarce. When the sub-GHz transmissions are used in order to increase the coverage, the data rate is usually very low. For all these reasons, decreasing the time-on-air of each transmission can reduce collisions, save bandwidth, and allow a higher number of coexisting nodes (for high-density applications [25]).

Therefore, it is clear that the new trend is to add “intelligence” to IoT devices so that they process data and make decisions without passing all the raw data to the cloud [9,26]. In this scenario, data processing is pushed to the “edge” of the network, even to IoT devices and sensors used primarily for data acquisition [27,28,29]. Thus, to enable services such as those mentioned previously, some solutions are available in the literature, such as [11,21,30], which proposed different data compression algorithms.

However, in those proposals, the algorithms were not structured with Machine Learning (ML) techniques, which is the conjunction of methods for data analysis that automates the construction of analytical models [31]. Machine learning algorithms allow the discovery of interesting patterns in the data, most of which are beyond what is possible by simple manual inspection. The combination of IoT devices and ML algorithms allows for a wide range of intelligent applications and enhanced user experiences, as shown in Figure 1.

This trend that is enabling the deployment of ML models in tiny, low-power, and low-latency IoT devices is called TinyML [7,32]. This alternative paradigm of TinyML tries to bring more intelligence to IoT devices, enabling the creation of novel applications that embed ML tasks in them. Furthermore, by allowing some data analysis and interpretation to be performed locally and in real time at the collection point, services as these can translate into huge cost savings and better privacy protection [7].

Most of the work performed in the field of TinyML has been focused on the reduction and optimization of existing models, such as Artificial Neural Networks (ANNs), to fit into these tiny devices and commodity microcontrollers, despite their computational restrictions [7,32,33]. Additionally, for IoT scenarios, it can be argued that the algorithms should preferably work without prior knowledge of the data, i.e., unsupervised. These are a class of models that are not explicitly instructed to predict something specific; rather, the patterns in the data are identified by the algorithm itself, and it is left to the user to interpret the output and reach a conclusion [34,35].

However, many of these existing models are static and require training phases to be fit to existing data. In contrast, in many IoT applications, the monitored systems are constantly evolving in real time and may encounter unpredicted disturbances and even complete concept drifts, where the characteristics of the data stream change [36]. In these cases, models designed offline might require constant recalibration. With this in mind, the concept of evolving algorithms can be used. These are a class of intelligent methods that can be defined as autonomous and data-driven, with the ability to evolve both their structure and internal parameters dynamically according to the new data received [36].

Thus, this work aimed to propose the development of a novel evolving algorithm for online data compression in IoT devices. The model, called Tiny Anomaly Compressor (TAC), was built from the ground up based on the Typicality and Eccentricity Data Analytics (TEDA) mathematical framework. As such, it does not require the use of previously established models or any assumptions about the underlying distribution of the data. Additionally, it uses recursive equations, which enable an efficient computation with a low computational cost, using little memory and processing power [37]. With all of these characteristics, the proposed TAC algorithm seems a perfect fit for IoT applications, as it constantly evolves over time as new data become available.

Another original work contribution is the analysis of a real-world case study used as a benchmark test for evaluating the proposed solution performance. In particular, a comparison with two other well-documented strategies was carried out, Swing Door Trending (SDT) and the Discrete Cosine Transform (DCT), respectively, as they are very well documented in the literature. The comparative analysis focused on evaluating the compression performance and error rates achieved by each of the models.

Finally, the contributions of this work are detailed below:The development of the novel TAC evolving algorithm;The methodology for the auto-definition of one of the hyperparameters of the TAC model;The performance benchmark with other well-known algorithms for validation;The definition of a potential new metric for time series compression evaluation, the Compression $F_{\beta }$ score (or $CF_{\beta }$).

The remainder of this paper is organized as follows: Section 2 details some concepts of data compression and provides a brief description of the SDT and DCT algorithms. Then, in Section 3, a few related works are presented. Following that, in Section 4, the mathematical basis for the TEDA metrics’ calculations is presented, whereas in Section 5, the proposed TAC algorithm is detailed. Afterwards, the methodology for the case study and benchmark evaluation are described in Section 6. Results are highlighted in Section 7. In Section 8, we detail the threats to the validity of our study. Finally, conclusions and future directions are indicated in Section 9.

## 2. Data Compression

Data compression can be characterized as the transformation or encoding of a set of symbols into another with a reduced size [9,11,30]. In information and code theory, data compression is one of the most relevant direct applications [17,38]. By definition, it seeks to detect redundancy in the data and tries to remove it through reduced representation. This approach is commonly accomplished by using compact representations for data that are more repeated and other longer ones for those with a lower frequency [17].

Such redundancies can be: (1) character distribution: within the domain or input alphabet, it is common for some symbols to occur more frequently than others; (2) repetition of characters: it is common for multiple repetitions of a character to appear; this sequence can be replaced by a single appearance of the character and a value indicating the number of repetitions; (3) redundancy: the frequency of data referring to the position within the input file; (4) patterns: the frequency of patterns is similar to the distribution of characters, but for a sequence of characters or symbols.

Therefore, according to [39], there are many universal compression techniques, such as (static or adaptive) Huffman and arithmetic coding, which are ubiquitous in real-world applications. In addition, it is known that they can be categorized into two types: lossy compression and lossless compression. In the first approach (lossy compression), data from the original file cannot be fully recovered after compression. As a result, the file size is permanently reduced, eliminating redundant data. In contrast, in the other approach (lossless compression), all original data from the compressed file can be recovered after decompression [40].

Alternatively, more domain-specific compression algorithms for different types of data began to emerge, such as text, image, audio, video [17], and, more recently, for IoT [40]. In this paper, the focus was on IoT and sensor time series data. For this type of scenario, lossy compression algorithms appear to generally achieve better compression efficiency, taking advantage of the existence of redundant information, as there is no need to recover it later [40].

It is important to note that some authors, as in [10], make a distinction between methods that rely on the different data encoding and transformation techniques to achieve compression and ones that adjust the sampling rate of the device dynamically, describing them as sampling optimization solutions rather than compression.

The end result of both types of algorithms tends to be the same: reducing the data storage footprint. Furthermore, the technique proposed in this paper does not fit nicely in either of those categories, as it does not transform the data, nor does it adjust the sampling rate in an online manner. Rather, the algorithm chooses which of the captured points are relevant to keep, discarding the rest. That being the case, we opted to use the terms interchangeably throughout this paper.

Finally, to test the newly proposed algorithm, called Tiny Anomaly Compressor (TAC), we used two well-documented compression algorithms found in the literature to perform a basic benchmark applied to a case study. The chosen algorithms were the DCT and SDT. These algorithms were selected due to being well documented in the literature and having a prolific use in compression tasks related to IoT scenarios, such as seen in [11,41], respectively. With all of these characteristics, they are suitable to provide a base bench-marking test for the TAC algorithm.

The DCT is a transform algorithm, similar to the discrete Fourier transform. The logic behind the use of transform techniques is to represent the data in a different domain, where they might be more easily compressible. The other fundamental aspect of the DCT is that the majority of the original signal energy is carried by only a few of the returned coefficients. This way, it is possible to discard upwards of 90% percent of the coefficients and still achieve a good representation of the original data after the decompression process (inverse DCT on the remaining coefficients) [41]. The main disadvantage of the DCT in relation to the other algorithms presented here is the fact that it needs to be executed offline with the complete dataset or in smaller batches. It does not work online and from a point-by-point analysis perspective.

In turn, the SDT algorithm is a compression method based on the concept of timed windows that uses linear trends to classify samples. The algorithm creates an area of coverage in the shape of a parallelogram between the last stored value and the most recent value [42], which it uses to classify the samples as relevant or not: samples that fall inside a window are considered similar to the original sample, thus potentially redundant, and can be discarded; on the other hand, if the sample falls outside of the window, it is classified as novel and saved [43]. Every time a new sample is stored, the algorithm creates a new window based on the sample value and the selected parameters [11].

It is important to keep in mind that, due to its popularity, several alternative implementations of the SDT exist in the literature, such as the ones proposed in [11,44,45]. Each of them suggests different improvements and changes to the original algorithm. That being the case, as the focus of the tests was to create a base benchmark, we chose to focus on the original version of the SDT for practical reasons.

Both of these algorithms, as well as the newly proposed TAC, are instances of lossy compression techniques, meaning that the original data cannot be fully reconstructed from the compressed data, as there is some information loss during the compression process. However, the goal of lossy compression algorithms is to preserve the maximum amount of essential information while getting rid of potentially useless data, thus reducing the overall storage requirements. Additionally, the compression process might also have a noise reduction effect, as the algorithms tend to filter out noisy samples, resulting in a, probably, smoother curve.

Therefore, as some portion of the information is lost due to the compression, there is always a trade-off between the amount of compression and the precision of the resulting data. For most applications, there must be a careful tuning of the model hyperparameters to achieve a good compression result while keeping the overall error within an acceptable range.

## 3. Related Works

A methodical search was conducted through the literature. As a result, several papers were found that impacted the direction of the research carried out. Therefore, eight of these articles were selected that contributed to the development of the proposed solution.

The SDT has been widely used for data compression in IoT scenarios. However, the main drawback is the fact that the SDT algorithm relies on a main user-defined parameter, the Compression Deviation (CD) [43]. The definition of an optimal value for this parameter is application dependent and requires testing or prior knowledge of the data behavior. Many authors proposed improvements and new auto-configuration steps to automatically set the CD parameter. In [45], the authors proposed the use of the Exponential Moving Average (EMA) as a means to define the CD in what they called Adaptive Swing Door Trending (ASDT). However, this function also requires the previous definition of the parameters.

In contrast, the improved SDT solution proposed in [11], called Self-defining SDT (SSDT), requires no prior parameters to be set. However, to do so, the algorithm needs to be trained on a subset of samples at the start of the applications. Their experiments showed good results, but, for other applications where sensor data can experience constant concept shift, it might not be as effective or else need to be periodically retrained.

On another front, the authors in [41] made use of transform-based lossy compression algorithms to compress the data from IoT weather stations. These algorithms work by transforming the data from the time domain to another domain where they can potentially be easier to compress. The three algorithms chosen were the Discrete Cosine Transform (DCT), the Fast Walsh–Hadamard Transform (FWHT), and the Discrete Wavelet Transform (DWT), from which the DCT was shown to perform best. The main drawback of this approach stems from the fact that data transformations cannot be performed online and on a point-by-point basis, as they need the full set of samples to be calculated.

In the work of [46], the authors proposed an online algorithm for streaming data compression, which took into account the generally concave trend of the compression ratio curve and optimized the key operation parameters (block size) through this metric. The algorithm successfully adapted one of the key parameters to the optimal value and yielded near-maximum compression ratios for time series data. The algorithm used was a variation of the Implementation of Dynamic Extensible Adaptive Locally Exchangeable Measures (IDEALEM), a lossy compression method based on statistical similarity. One of the disadvantages of this approach is that a number of data blocks need to be saved to be later statistically compared to newer samples.

In the approach of [10], a multi-layer efficient data transmission strategy for IoT monitoring systems was proposed. The proposed solution was based primarily on two mechanisms. First, they proposed to gather data from multiple sensor nodes into a gateway node. There, essential Key Performance Indicators (KPIs) were collected from the data to be analyzed. The KPIs were defined according to specialists and used to distinguish between important information among all the data. This way, essential features in the data can be sent in real time to the cloud server, while the rest are stored in a local buffer to be periodically transmitted. The second mechanism was the use of a standard zip compression algorithm on these stored data to reduce the amount of data to be transmitted to the cloud. This approach was designed in such a way so as to ensure both the immediate delivery of critical information and the delayed delivery of detailed information with network resource awareness for other messages. The disadvantage came from the need to define the KPIs, as they were very application-dependent, so prior knowledge was needed.

In [47], they proposed to compress industrial data using neural network regression into a representative vector with lossy compression. For the efficiency of the compression, they used the divide-and-conquer method, dividing the data by time and using machine learning technology in the conquer process. Furthermore, with additional techniques, they tried to randomly select a value around the range for conquering, but it was not affected. The drawback of the proposed techniques was that none of them could be performed in an online manner.

In the work [9], they proposed to adapt the SZ algorithm for IoT devices by considering only the floating-point data type, which makes the code smaller and easier to compile on resource-constrained devices. Additionally, the algorithm was adapted to take a 1D array of float sensor data as input and return a byte array to be transmitted to the edge node. In this method, the data are stored locally and transmitted to the edge after each period *P* of time. Compared to the proposed work, the main difference was that the compression process transformed the data into a byte array, which later needed to be decompressed before they became usable.

In the study [39], the author proposed a two-level compression model that selects a proper compression scheme for each individual point, making it possible for diverse patterns to be captured with fine granularity. Based on this model, they designed and implemented the Adaptive Multi-Model Middle-Out (AMMMO) framework, which provides access to a set of control parameters to categorize data patterns. However, to effectively handle diverse data patterns, they introduced a reinforcement learning-based approach to learn parameter values automatically. The main disadvantage of this approach is its complexity, making it possible to only be deployed at the database level. That being the case, it cannot be applied directly to an online IoT scenario.

Finally, in Table 1, a comparative summary of the characteristics of the works mentioned in this section is presented. There, it is possible to visualize their main similarities and differences. Each row represents one particular work, with the characteristics of the proposed work being presented on the last row. Finally, each of the four columns indicates topics (or features) and how they were covered in the given research. The four features analyzed here were: if the algorithm was executed online; if the proposal used machine learning for compression; if the solution was lightweight; and if the compressed data were encoded in a different format to achieve compression.

Thus, it is clear in the literature that studies that developed compression algorithms using machine learning are still few. Furthermore, none of them were found to be lightweight and online solutions. Therefore, from the discussion presented above, it is clear that there are still gaps to be explored in this area, which favors the development of new solutions, for data compressing for the IoT, mainly for the emerging concept of TinyML.

## 4. Typicality and Eccentricity Data Analytics

The Typicality and Eccentricity Data Analytics (TEDA) mathematical framework, sometimes also called Empirical Data Analysis (EDA) in the literature, was initially proposed in [48] and later expanded in a series of other articles and the book Empirical Approach to Machine Learning [36].

This new set of concepts comes as an alternative to traditional machine learning approaches that are mathematically based on traditional probability theory. TEDA, as its name suggests, is based on the use of the concepts of eccentricity and typicality. These two quantities represent, respectively, the density and proximity of the samples in the data space [48].

This new approach is also statistical, but it differs from the traditional probability theory, which is, in its nature, frequentist and ideal for describing purely random processes [48]. In fact, despite its solid theoretical basis and widespread use today, the theory of probability was developed with purely random processes in mind, such as rolling dice or currency. Thus, it is perfectly suited to describe such purely random processes and variables [37].

However, the biggest motivation for the development of this new methodology is the fact that real processes are, for the most part, not purely random [48]. Thus, unlike random processes, real processes violate the main premises that the traditional probability theory requires [48].

As a result, the TEDA framework emerges as an alternative systematic methodology that does not require, among other things: previous assumptions about the data distribution; the independence of individual data samples (observations); an infinite number of observations, being able to work with only three data samples [37,48]. Therefore, because it is based entirely on the data and its mutual distribution in the data space, the approach can be categorized as empirical in nature.

In the following subsections, the main mathematical concepts behind the methodology are shown and what makes it so flexible that it can be used for the development of a variety of algorithms, such as: anomalies and failure detection [49], image processing, clustering [50,51], classification [52], regression [53], forecast, control, filtering, big data [54], and traffic analysis [55], among others.

### 4.1. Metrics’ Calculation

TEDA is based on several quantities that are calculated from the proximity/similarity of the samples in the data space. However, these measurements are not exactly equal to the density used in statistics and other areas [37].

Among the new concepts introduced by TEDA, three of them are detailed here: accumulated proximity (π), typicality (τ), and eccentricity (ξ).

We start by defining the $x\in R^{n}$ data space, where **x** represents the data stream in an *n-dimensional* space. In this case, the ordered sequence of received data can be represented as follows:(1)$$\begin{array}{c} \left\{x_{1},x_{2},\ldots ,x_{k},\ldots \right\},\;x_{k}\in R^{n},\;k\in N \end{array}$$

Thus, we have that each sample $x_{k}$ is a vector of *n* dimensions that represent the system in the discrete instant *k*. The methodology can be extended to samples of any dimensionality *n*. However, as an example, let us say that we want to observe a phenomenon in which samples are described in a space of one dimension, such as the temperature of a thermostat, for example.

In the context of a data stream, samples can be described as being received discreetly at times *k*, the first sample being received when k=1, the second when k=2, and so on. Therefore, for any set of two or more samples (k≥2), we can define a distance metric between them, d(x,y), whether it is the Euclidean, Mahalanobis, Manhattan, cosine, or other [48].

From this, we have that the accumulated proximity (π) of a point *x* is calculated by adding the distances between the point *x* and all the *k* points of the set. This calculation can be seen in Equation (Equation 2) [48].
(2)$$\begin{array}{c} \pi_{k}\left(x\right)=\sum_{i=1}^{k}d\left(x,x_{i}\right),\;k\geq 2 \end{array}$$

The values of typicality (τ) and eccentricity (ξ) can be defined as soon as three or more different samples have been obtained (k≥3). This is due to the fact that any pair of non-identical samples is equally atypical/eccentric [48]. In this manner, the eccentricity of a sample *x* in an instant *k* is defined by the ratio between its accumulated proximity and the sum of the accumulated proximities of all other samples, as seen in Equation (Equation 3) [48].
(3)$$\begin{array}{c} \xi_{k}\left(x\right)=\frac{2\pi_{k}\left(x\right)}{\sum_{i=1}^{k}\pi_{k}\left(x_{i}\right)},\;k>2,\;\sum_{i=1}^{k}\pi_{k}\left(x_{i}\right)>0 \end{array}$$

The typicality (τ), in turn, is defined as the complement of the eccentricity, shown in Equation (Equation 4).
(4)$$\begin{array}{c} \tau_{k}\left(x\right)=1-\xi_{k}\left(x\right) \end{array}$$

It is very important to note that the functions of eccentricity and typicality are bounded and, therefore, have normalized versions [53]. Those are presented here in Equations (Equation 6) and (Equation 8), respectively.
(5)$$\begin{array}{c} 0\leq \xi_{k}\left(x\right)\leq 1,\;\;\sum_{i=1}^{k}\xi_{k}\left(x_{i}\right)=2 \end{array}$$
(6)$$\begin{array}{c} \zeta_{k}\left(x\right)=\frac{\xi_{k}\left(x\right)}{2} \end{array}$$
(7)$$\begin{array}{c} 0\leq \tau_{k}\left(x\right)\leq 1,\;\;\sum_{i=1}^{k}\tau_{k}\left(x_{i}\right)=k-2 \end{array}$$
(8)$$\begin{array}{c} t_{k}\left(x\right)=\frac{\tau_{k}\left(x\right)}{k-2} \end{array}$$

These two functions are very important, as they behave similarly to the Probability Density Function (PDF) of traditional probability theory. However, unlike the PDF function, they do not require any prior knowledge about the data distribution and represent both the spatial distribution of the samples and their frequency of occurrence [48]. The typicality function also resembles histograms of probability distributions and fuzzy membership functions [37].

### 4.2. Recursive Case

The calculations shown so far are applicable to online data streams when samples are being received in real time and the value of *k* is continuously increased. They are also applicable to previously stored datasets when it is desired to use the algorithm offline.

However, all of these definitions are global, i.e., it is necessary to have access to the entire dataset prior to the application of the algorithm (if offline) or to keep a local record of all samples that have already been received in the past (online case). Thus, its use in IoT scenarios and in edge computing devices is impractical because, as discussed in the previous sections, there are major limitations of data storage and processing power on these devices.

In this case, the recursive method of calculating the values of typicality and eccentricity is particularly useful. The mathematical proof of these calculations was presented for Euclidean and Mahalanobis distance in [52]. With this method, it is not necessary to keep previous samples stored locally; instead, the calculation of the metrics can be performed using only the last sample received and aggregate values that represent the state of the system at the previous instant (k−1).

This can be obtained by calculating the mean (μ) and the variance ($\sigma^{2}$) also in a recursive manner for each new sample received. This is shown in Equations (Equation 9) and (Equation 10) [37].
(9)$$\begin{array}{c} \mu_{k}\left(x\right)=\frac{k-1}{k}\mu_{k-1}+\frac{1}{k}x_{k},\;\;\mu_{1}=x_{1} \end{array}$$
(10)$$\begin{array}{c} \sigma_{k}^{2}\left(x\right)=\frac{k-1}{k}\sigma_{k-1}^{2}+\frac{1}{k-1}{∥x_{k}-\mu_{k}∥}^{2},\;\;\sigma_{1}^{2}=0 \end{array}$$

With this, it is possible to use the recursive format of the equations to calculate the eccentricity and typicality, presented in Equations (Equation 11) and (Equation 12), respectively [37].
(11)$$\begin{array}{c} \xi_{k}\left(x\right)=\frac{1}{k}+\frac{{\left(\mu_{k}-x_{k}\right)}^{T}\left(\mu_{k}-x_{k}\right)}{k\sigma_{k}^{2}} \end{array}$$
(12)$$\begin{array}{c} \tau_{k}\left(x\right)=1-\xi_{k}\left(x\right)=\frac{k-1}{k}-\frac{{\left(\mu_{k}-x_{k}\right)}^{T}\left(\mu_{k}-x_{k}\right)}{k\sigma_{k}^{2}} \end{array}$$

This recursive method used to update the values of eccentricity and typicality makes the execution of the algorithm very computationally efficient. In addition, even without the previous samples having to be stored locally, the calculations are performed in an exact manner and not by some approximation. As a result, there is no loss of precision in the process of updating the metrics [52]. Thus, the TEDA metrics’ calculations are ideal for applications with edge computing systems and real-time processes that have major limitations in computational resources and data storage.

### 4.3. Anomaly Detection

The Chebyshev inequality is a statistical principle widely used to detect anomalies. Assuming a sufficiently large set of samples from any data distribution, Chebyshev’s inequality ensures that no more than $1/m^{2}$ samples will be more than an mσ distance away from the mean [56].

In [37], the authors demonstrated that it is possible to derive a condition that provides exactly the same result (but without making assumptions about the amount of data, their independence, and so on) as Chebyshev’s inequality, but represented in terms of TEDA. This new inequality can be seen in Equation (Equation 13).
(13)$$\begin{array}{c} \zeta_{k}\left(x\right)>\frac{m^{2}+1}{2k} \end{array}$$

This formula, called the eccentricity inequality [53], allows local and individual analyses to be carried out for each new sample obtained in an online data stream. For this, it is only a matter of calculating the normalized eccentricity of every new sample, and if it exceeds the limit, the sample is classified as an anomaly.

## 5. The Proposed Algorithm

With the theory presented in Section 4 in mind, this paper aimed to propose the development of a novel evolving algorithm to perform local and online data compression on smart, IoT-enabled devices. This new algorithm is called the Tiny Anomaly Compressor (TAC).

Being built on top of the concepts of typicality and eccentricity presented earlier, the algorithm does not require previously established mathematical models or any prior assumptions about the data distribution. In addition, it is based on recursive calculations, which makes it an efficient algorithm, with low computational cost and low memory usage and processing power [37]. The only requirement for the algorithm to perform well is that the time series samples present some degree of spatial or time correlation among each other, which is exactly the case for the data stream generated by a sensor measuring some variable over time.

In this section, the basis for the established TEDA anomaly detection process is initially presented. As the algorithm was developed using this process, it is important to show how it works before presenting the full compression algorithm. Then, the logic behind the TAC algorithm and its functionality are shown, along with its pseudocode and basic execution flow diagram.

### 5.1. Anomaly Detection with TEDA

The general sequence of steps necessary to perform the anomaly detection with TEDA are as follows: for each new sample received, the internal mean and variance metrics are updated, then the sample eccentricity is calculated, and making use of the eccentricity inequality presented in Equation (Equation 13), it is classified as an anomaly or not.

The only hyperparameter needed is the value of *m* to be used in the eccentricity inequality evaluation. This hyperparameter can be interpreted as the sensitivity of the model to the presence of anomalies. From this base structure, it is possible to derive a series of more specialized algorithms, such as clustering [50] and, now, a lossy compression one defined in this work.

### 5.2. TAC Algorithm

The TAC algorithm, as the name suggests, makes use of the anomaly detection process to perform the time series compression. However, there are some peculiarities in relation to the basic structure presented above.

The first of these differences is the introduction of the concept of a rolling and dynamic “anomaly window”. To do this, we introduced a counter variable that counts how many anomalies have been found since the last saved sample. To control the behavior of these windows, a new hyperparameter needs to be introduced: the window_limit. With it, the user can control the number of anomalies that should be detected before the window is considered “full”.

The big difference of the algorithm execution arises as a result of this concept of the “window of anomalies”. This happens because, whenever the window is not yet full, no new samples should be saved. However, once the window is filled and a new anomaly is detected, this sample should be saved. At this step, the model is said to be detecting a possible concept drift in the series. At this point, the window should be reset and the model’s internal parameters (*k*, mean, and variance) returned to their initial values to start a new window.

The window is said to be dynamic because it has no fixed size in regard to the total number of data samples included in them. The window is only controlled by the number of anomalies detected. Therefore, the total number of samples in each of them is determined by the intrinsic variability of the dataset.

The *window_limit* hyperparameter can be interpreted as the response delay of the model to a concept drift in the signal. For higher values of *window_limit*, for example, the model has to detect a larger amount of anomalous data points to trigger an internal parameter reset, thus detecting a concept drift and saving a new sample.

Finally, the pseudocode of the TAC algorithm is given in Algorithm 1.

### 5.3. Auto-Definition of the m Hyperparameter

Another difference introduced to the TAC algorithm is the ability to dynamically auto-define the *m* hyperparameter if the user chooses to do so. This possibility leaves the model with only one controllable hyperparameter (the *window_limit*) to be set. To implement this feature, another concept of traditional probability theory was borrowed: the calculation of the kurtosis of the distribution.

In traditional probability theory, the kurtosis is a measure of the tendency of the distribution to contain anomalies (long tail). It is a scaled version of the distribution’s forth moment [57]. A similar metric could be calculated from the recursive TEDA variables, as shown in Equation (Equation 15).
(14)$$\begin{array}{c} fourthMoment_{k}\left(x\right)=\frac{k-1}{k}\;fourthMoment_{k-1}+\frac{1}{k-1}{∥x_{k}-\mu_{k}∥}^{4} \end{array}$$
(15)$$\begin{array}{c} kurtosis_{k}\left(x\right)=\frac{fourthMoment_{k}}{\sigma_{k}^{4}} \end{array}$$

In early experimental testing, it was found that the performance of the algorithm would degrade quickly for values of *m* greater than 1.2. As a result, it was chosen to determine its value following the condition shown in Equation (Equation 16).
(16)$$\begin{array}{c} m=\left\{\begin{array}{cc} |kurtosis_{k}| & \mathrm{if}\;|kurtosis_{k}|\leq 1\\ \frac{1}{|kurtosis_{k}|} & \mathrm{otherwise} \end{array}\right. \end{array}$$

Finally, the value of *m* can be recursively calculated for each new sample before the evaluation with the eccentricity inequality.
**Algorithm 1:** Tiny anomaly compressor.  **input**  **:** Data stream *X*, Float *m*, Integer window_limit  **output****:** keep_point  **1**
**procedure** TAC (*Data stream X, Integer window_limit*)  **2**  **while** *X is active* **do**  **3**     read sample $x_{k}\in X$  **4**     **if** k==1 **then**  **5**     $\mu_{k}=x_{k}$  **6**     ${\left[\sigma^{2}\right]}_{k}^{x}=0$  **7**     **if** time==1 **then**  **8**         keep_point = true  **9**     **else****10**         keep_point = false**11**     **end****12**     **else****13**     $\mu_{k}=\frac{k-1}{k}\mu_{k-1}^{x}+\frac{x_{k}}{k}$**14**     ${\left[\sigma^{2}\right]}_{k}^{x}=\frac{k-1}{k}{\left[\sigma^{2}\right]}_{k-1}^{x}+\frac{∥x_{k}-\mu_{k}{∥}^{2}}{k-1}$**15**     **if** *x==last_value* ***and**** $\sigma^{2}==0$* **then****16**         keep_point = false**17**     **else****18**         $\xi_{k}\left(x\right)=\frac{1}{k}+\frac{{\left(\mu_{k}^{x}-x_{k}\right)}^{T}\left(\mu_{k}^{x}-x_{k}\right)}{k{\left[\sigma^{2}\right]}_{k}^{x}}$**19**         $\zeta_{k}\left(x\right)=\frac{\xi_{k}\left(x\right)}{2}$**20**         **if** $\zeta_{k}\left(x\right)>\frac{m^{2}+1}{2k}$ **then****21**         is_anomaly = true**22**         anomaly_count = anomaly_count + 1**23**         **else****24**         is_anomaly = false**25**         **end****26**         **if** anomaly_count>=window_limit **then****27**         resetWindow()**28**         keep_point = true**29**         **else****30**         keep_point = false**31**         **end****32**     **end****33**     **end****34**     time = time + 1**35**     k = k + 1**36**     last_point = x**37**  **end**

### 5.4. Execution Flowchart

As a means of clearly demonstrating the execution steps of the TAC model when inserted into a real scenario, two flowcharts can be observed in Figure 2. In Figure 2a, a general representation of the data acquisition and transmission process carried out by a microcontroller without the use of a compression algorithm can be seen. In contrast, in Figure 2b, it is possible to see the additional steps performed when the TAC model was deployed.

## 6. Case Study Definition

The case study aimed to evaluate the impact of the three different lossy compression algorithms and the fidelity of the recovered data after restoration/decompression. With this, we aimed to validate the newly proposed TAC algorithm. This section presents the benchmark test that was developed, with a focus on the dataset selection, objective definition, evaluation metrics, and execution process.

### 6.1. Dataset Selection

The datasets used in this work were obtained from a huge experiment currently ongoing at the eLUX laboratory at the University of Brescia, Italy. eLUX is a living laboratory for smart energy, smart grids, and smart environments for Industry 4.0 [58]. The laboratory concentrates and analyzes data coming from many distributed plants and from a huge quantity of sensors, meters, and actuators of different types, implementing an excellent example of the smart grid based on industrial-grade IoT systems. The permanent data storage and the specific web services made all the information available to researchers and other authorized users.

In this work, one of the use cases of the eLUX laboratory was considered. Since 2017, eLUX has run the energy supervision and management system of the entire Engineering Campus [59]. In particular, the data coming from one of the (four) Photovoltaic (PV) plants of the campus were used. This PV plant is powered by a string of 15 PV panels in series, whose current and voltage are monitored by innovative IoT meters with a sampling interval of 5s. Please note that the usual sampling time of standard commercial meters is 15 min, while in the eLUX experiments, a 90× oversampling is used. Such a high sampling rate enables the use of innovative performance analysis algorithms and predictive maintenance (based on neural network processing). When this approach is scaled to the entire campus (or an entire smart city), such a high sampling rate may generate huge quantities of data, requiring efficient compressing algorithms, as the one proposed in this paper.

The first dataset was obtained by extracting 50,879 values from the database of the PV string voltage, while the second dataset included the corresponding 50,879 values of the PV string current. Further details about the data statistical properties can be seen in Table 2.

### 6.2. Objective Definition

The main goal was to conduct a base benchmark test to analyze the performance of the newly developed TAC data compression algorithm for IoT sensor data streams.

For further validation, two other algorithms found in the literature were used for comparison: SDT and the DCT. The test targeted assessing the performance differences of the three algorithms to compress the continuous data streams presented before in a simulated scenario. In this context, we aimed to find the maximum compression rate that could be achieved by the algorithms and which of them had a better overall performance in the analyzed scenario.

### 6.3. Evaluation Metrics

To evaluate the compression results of the tests, several metrics were chosen. This was necessary as the lossy compression process needs to be evaluated in regard to both the resulting Compression Error (CE), as well as the achieved Compression Rate (CR). As the performance on these two fronts tends to be inversely proportional, both of them had to be considered in the evaluation.

Furthermore, in order to highlight the different characteristics of each of the algorithms, the extra metrics of the Peak-Signal-to-Noise Ratio (PSNR) and the Normalized Cross-Correlation (NCC) were selected. Finally, to perform the hyperparameter optimization, a novel metric was introduced that took into account both the CR and CE of the models.

The definitions of all these metrics are as follows.

#### 6.3.1. Compression Error

The compression error measures the difference observed between the original time series and the reconstructed data after the compression and decompression processes. To generate the decompressed series for the TAC and SDT compressor, simple linear interpolation was performed to fill the gaps between the saved samples. For the DCT compressor, on the other hand, the inverse DCT transformation was applied to the saved coefficients to obtain an approximation of the original signal.

To measure the error between the original signal and the decompressed signal, the primary metrics chosen were the Mean Absolute Error (MAE) and the Mean-Squared Error (MSE), which are shown in Equations (Equation 17) and (Equation 18).
(17)$$\begin{array}{c} MAE=\frac{1}{n}\sum_{i=1}^{N}|x_{i}-{\hat{x}}_{i}| \end{array}$$
(18)$$MSE=\frac{1}{n}\sum_{i=1}^{N}(x_{i}-{\hat{x}}_{i}{)}^{2}$$
where $x_{i}$ are the samples of the original signal, ${\hat{x}}_{i}$ are the samples of the decompressed signal, and *n* is the total number of samples.

#### 6.3.2. Compression Rate

As is the case for measuring the error, there are several approaches to quantify the effectiveness of the compression on the size reduction. For consistency across the algorithms, the CR was measured as the percentage of size reduction between the original file and the file with the compressed signal representation. This value is calculated as shown in Equation (Equation 19).
(19)$$\begin{array}{c} CR=\left(1-\frac{size(compressedFile)}{size(originalFile)}\right)\times 100 \end{array}$$
where size() is the function that returns the file size in KB.

To keep everything consistent, after every test, both the original and the generated compressed time series were saved in a standardized form as different CSV files. Each file only contained two columns: one for the measured sensor value on the given timestamps and another for the respective sample indices (used for the data reconstruction with linear interpolation). For the special DCT case, the selected coefficients returned by the model were saved instead of raw sample values.

#### 6.3.3. Peak-Signal-to-Noise Ratio

The PSNR was also calculated. The PSRN is a metric used to evaluate the effect of noise in the signal representation. Its a commonly applied technique in fields such as digital image processing, where it can be used to measure the quality of images after passing through noise reduction filters [60].

The PSNR is presented in decibels (dB) using a logarithmic scale, and usually, higher values mean less presence of noise and, thus, a better representation. It can be used to highlight the denoising capability of the models and is calculated as shown in Equation (Equation 20).
(20)$$\begin{array}{c} PSNR=10\times \log_{10}\left(\frac{{\left(max\left(x\right)-min\left(x\right)\right)}^{2}}{MSE(x,\hat{x})}\right) \end{array}$$
where MSE() is the mean-squared error and max() and min() are functions that return the maximum and minimum values of the original signal, respectively.

#### 6.3.4. NCC

The Normalized Cross-Correlation (NCC) coefficient is a measure of the similarity between two signals. It is another metric common in the fields of signal and image processing [61,62]. Its value is limited to between negative one and one. As a result, in a perfect scenario where the decompressed signal would be exactly the same as the original, for example, their NCC would be exactly one. It can be calculated as shown in Equation (Equation 21).
(21)$$\begin{array}{c} NCC=\frac{1}{n}\sum_{i=1}^{N}\frac{\left(x_{i}-\mu_{x}\right)\left({\hat{x}}_{i}-\mu_{\hat{x}}\right)}{\sigma_{x}*\sigma_{\hat{x}}} \end{array}$$

### 6.4. Proposed CF Score

For a balanced operation, both the CR and CE should be considered when selecting optimal compressor hyperparameters. With that in mind, we proposed a novel metric inspired by the $F_{\beta }$ score for statistical analysis.

In traditional probability theory, the $F_{\beta }$ score is a weighted average metric between the precision and recall rates of a classifier [63]. The β in the $F_{\beta }$ score is a user-defined parameter that controls the importance of the rates in the calculation. Additionally, the metric computation is based on the harmonic mean formula. Therefore, when both rates have an equal weight (β = 1), the $F_{1}$ score reverts to being a simple harmonic mean between the two values used [63].

In turn, the harmonic mean is an average measure generally considered as a fusion technique of numerical data information [64]. It differs from the traditional arithmetic mean by giving more weight to smaller values being averaged and is preferred when averaging rates or ratios [65].

In the compression case, the CR is already a ratio between zero and one. However, the error magnitudes are highly dependent on the data scale; thus, it is not a normalized value. As a replacement, we proposed to use the NCC, and instead of trying to minimize the error, we tried to maximize the similarity between the signals.

With all that in mind, the so-called Compression $F_{\beta }$ score ($CF_{\beta }$ score) can be calculated as shown in Equation (Equation 22).
(22)$$\begin{array}{c} CF_{\beta }=\left(1+\beta^{2}\right)\frac{CR*NCC}{(\beta^{2}*CR)+NCC} \end{array}$$
where β is a positive real parameter used to adjust the importance of each metric in the $CF_{\beta }$ calculation. The parameter was chosen such that the NCC was β times more important than the CR. Finally, when β=1, the equation puts equal weight on both metrics and turns into a simple harmonic mean.

This proposed $CF_{\beta }$ score is somewhat similar to the Compression Criteria proposed in [11], where they proposed the use of a harmonic mean between the compression rate and error. However, $CF_{\beta }$ takes into account the preferred use of rates when calculating the harmonic mean by replacing the error with the NCC score. It also is inspired by the $F_{\beta }$ score and adds the β parameter to adjust the weight of the two metrics.

### 6.5. Design and Operation of the Benchmark Test

To perform the benchmark test, the newly developed TAC algorithm (described in detail in Section 5) was used alongside two other well-established time series compression algorithms found in the literature, SDT and the DCT, both of which were briefly described in Section 2. The test was carried out in an offline manner, as all the data were already previously acquired.

The first barrier encountered in the design of the test was the different hyperparameters available to tune in each of the models. Since they were not the same across all three models, it was not possible to fix a single value for all of them to standardize the tests. This being the case, the first step was to perform a grid-search on each of the models to find good hyperparameter values. This consisted of executing the model multiple times on the dataset with different combinations of hyperparameters. This was performed to find the optimal configuration for each of them on the respective datasets. With the ideal hyperparameter combination, the algorithms had the best “all around” performance on the data and should be more easily comparable to each other in a benchmark scenario.

The second barrier arose from the execution of the grid-search itself. When trying to optimize hyperparameters, a metric needs to be chosen to be minimized or maximized to assess the performance. However, in the case of lossy compression, there are two fundamental metrics that have to be taken into account, the CR and CE, and they are essentially inversely proportional. For example, if the parameters are chosen so that the CR is maximized, the resulting error will be high, and the result might lose too much information; inversely, if the CE is chosen to be minimized, the model will achieve poor compression results.

For this reason, it was chosen to develop and use the $CF_{\beta }$ score when searching for optimized hyperparameters. When trying to maximize this metric, both the compression rate and the similarity of the resulting signal were taken into account. In this manner, it was possible to find a more balanced result for each model performance. Additionally, the use of the $CF_{\beta }$ score on the grid-search enabled the use of different values of β to change the test configuration for all models at once.

After the grid-search was performed, the models were executed on the dataset with their best hyperparameters. Then, the evaluation metrics presented in the beginning of this section were applied to each of the results, generating a performance report for each model that was then compiled to be analyzed.

This entire process was performed for the two datasets described at the beginning of this section. Furthermore, to achieve more variability in the results, the entire process was repeated twice for each dataset, once using β=1 and once using β=4. With this, it was possible to observe two different behaviors for the models in each dataset. When β=1, both the compression rate and similarity were considered to have the same weight when finding optimal parameters. However, when β=4, the similarity was considered four-times more important than the compression rate, leading to results with less error.

In all of these rounds, the DCT and SDT models were executed once and the TAC model was executed twice: one with both hyperparameters being defined and one where the *m* hyperparameter was defined dynamically by the model itself. At the end of each round, the performance metrics of the algorithms were collected for each dataset and the different values of β that were chosen.

## 7. Results and Discussion

To present the results in a clear manner, it was decided to separate the discussion for each dataset used in the case study. First, the results regarding the voltage dataset are shown and, finally, the results for the current dataset. The first dataset consisted of 50,879 samples of voltage measurements, whereas the second dataset had the same amount of samples of electrical current measurements.

### 7.1. Dataset 1: Electrical Voltage

As stated earlier, the first step was to perform a grid-search to find the optimal hyperparameters for each algorithm. The process was performed by repeatedly running the models on the data with different hyperparameters, then choosing the configuration that yielded the best $CF_{\beta }$. As an example, in Figure 3, it is possible to see the result of the process graphically for the SDT and TAC (with dynamic *m*) models when β=1. The figure illustrates the variation of the compression rate, NCC, and CF score obtained when varying the hyperparameter for each of the models. It is also possible to see the point where the maximum value of the CF score was achieved, marked with a grey arrow.

As is possible to see, the CR values tended to increase following a logarithmic trend, while the NCC, on the other hand, decreased following a somewhat linear (albeit noisy) tendency. For this reason, it was also decided to test the models with a greater value of β, so that the NCC would have a larger effect on the $CF_{\beta }$ calculation and, thus, the hyperparameter definition.

In Table 3, it is possible to see the optimal hyperparameters found for every model for both β=1 and β=4.

It is possible to observe a good difference in the parameters for both the TAC and SDT models, but not so much in the DCT. This could hint at greater flexibility of the SDT and TAC models in relation to the DCT. Finally, with the optimal hyperparameters defined, the models were executed again with their respective configurations. The compression results for β=1 and β=4 can be seen in Table 4.

The TAC model (both with dynamic *m* and not) outperformed the two other models in regard to the total error and NCC in all cases. They also achieved greater values of the PSNR, indicating less noise in the decompressed signal. Additionally, for β=1, the TAC also achieved a greater compression rate than SDT, only losing to the DCT by a small margin. In the scenario where β=4, the TAC and SDT resulted in similar compression rates, again losing by a little to the DCT.

Additionally, in Figure 4, the original and decompressed signals are displayed. The original signal is plotted in red, and the resulting decompressed signal is on top of it in blue. In Figure 4a–d, it can be noted that the TAC model (both with dynamic *m* and not) was able to ignore most of the samples that were big outliers, reducing the signal noise. SDT, which uses linear trends to classify the samples, was not able to do the same. The DCT, in turn, was also able to ignore most of the clear outliers; however, it also seemed to introduce a little noise in sections where the signal should be smoother. This can be attributed to a byproduct of the signal reconstruction process by cosines when applying the inverse DCT.

Finally, in Figure 4e–h, it is possible to see that, when β=1 and the NCC were taken into account better, the TAC no longer removed the outliers, as it was more focused on maximizing the signals’ similarity. However, it was still able to achieve much less compression error than the other two, and without sacrificing too much of the compression rate.

### 7.2. Dataset 2: Electrical Current

The same sequence of steps was repeated for the second dataset. The optimal hyperparameters found for each model can be seen in Table 5.

The compression results for β=1 and β=4 can be seen in Table 6. It can be observed that very similar results to the first experiment were obtained, where the TAC model (both with dynamic *m* and not) consistently achieved lower values of the error than the two competitors.

Similar to the other experiment, the original and decompressed signals can be seen in Figure 5 for both β=1 and β=4, respectively.

## 8. Threats to Validity

The threats to validity for the present study were the following:Construct validity: The construction validity verifies the relationship between the theory behind the experiments carried out and the observations found. This threat is mainly related to the algorithms used in the experiment. Therefore, we were confident that the descriptions and pseudocodes found for the studied algorithms were correct since the results were satisfactory.Internal validity: Internal threats are relatively linked to the experiment since it is necessary to define different parameters for the algorithms. Thus, to mitigate this threat, a grid-search was performed on each of the models analyzed.External validity: External threats refer to the ability to generalize our findings and conclusions appropriately to other contexts. In this study, the experiments were carried out with large sets of data (voltage and current), which allowed us to analyze the compression. Although the tests were executed only on two datasets, we were confident that the results for other sets were similar.

## 9. Conclusions

This paper proposed an evolving, dynamic, and lightweight data compression algorithm for IoT environments. This is a fast-growing field, as applications with IoT features are growing in number exponentially, bringing with them new challenges due to the increase of both the connected sensors and the data generated by them. In order to achieve this objective, a benchmark test for the validation of the novel TAC algorithm was introduced. Moreover, two other algorithms from the literature (SDT and DCT) were used for comparative analysis, from which the results were shown in the last section.

The TAC model is based on data eccentricity and is “empirical” in nature, as it does not require any previous assumptions for the underlying data distribution or statistical properties and learns strictly from the data’s spatial and temporal relations. Furthermore, unlike the SDT model, which uses linear trends, the TAC model has no predefined shape and is adjusted as the continuous data stream progresses. On top of that, the TAC model is based on simple and recursive calculations, making it lightweight both in regard to the storage space and processing power needed for execution.

Regarding the maximum compression rate that can be achieved by the algorithms, the TAC model was able to reach a maximum CR of 97.49% with a 1.6 MAE in the first dataset and a maximum CR of 98.33% with a 0.1 MAE for the second dataset. On the other hand, SDT achieved a CR of 97.15% with a 2.33 MAE on the first dataset and a CR of 98.98% with a 0.26 MAE on the second. Finally, the DCT obtained a CR of 98.8% with a 4.2 MAE for the first dataset and a CR of 99.48% with a 0.18 MAE on the second.

For this reason, it was concluded that the best performing model in the context analyzed was the newly proposed TAC algorithm. It obtained comparable compression rates to the other two models, even beating them in some cases, while consistently managing to deliver less error on the output signal. Sometimes, the DCT attained slightly larger compression rates, but it cannot be executed online and in a point-by-point analysis scenario, as the TAC does.

In conclusion, the TAC algorithm is a novel and interesting contribution that deserves major attention.

Future work includes, but is not limited to: expanding the compression algorithm to work for multiple variables simultaneously; and evaluating the impact on the performance of the TAC algorithm on an embedded IoT device.

## Figures and Tables

**Figure 1 sensors-21-04153-f001:**
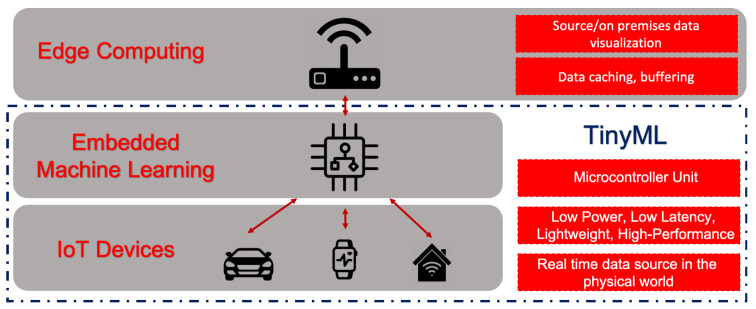
TinyML ecosystem-as-a-service and the use of machine learning at the edge.

**Figure 2 sensors-21-04153-f002:**
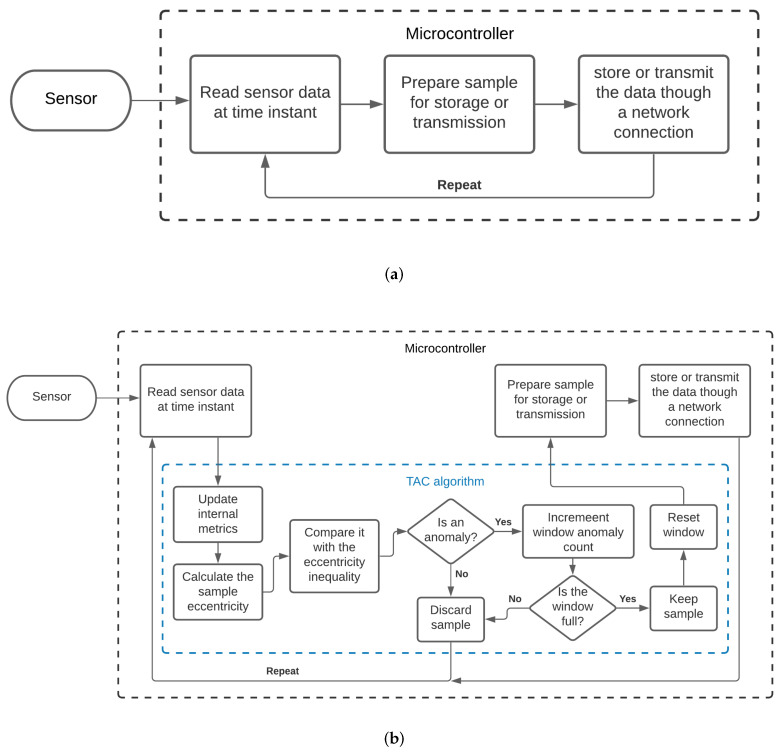
Execution flow. (**a**) Data flow without the TAC model; (**b**) Data flow with the TAC model.

**Figure 3 sensors-21-04153-f003:**
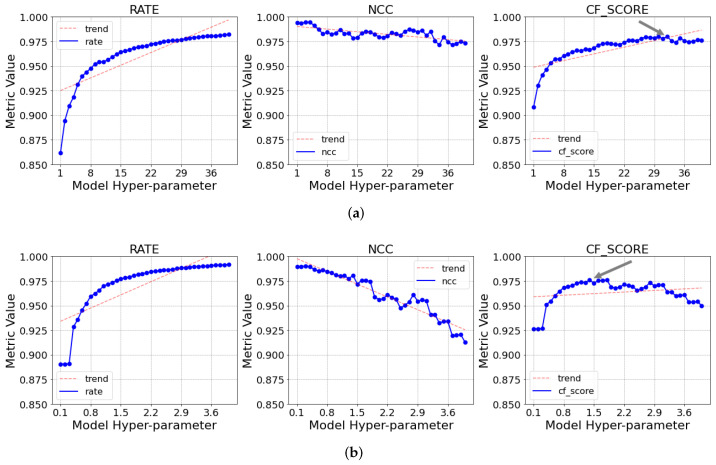
Variation of the compression rate, NCC, and CF score for the grid-search evaluated by varying the hyperparameter for the TAC and SDT models. β=1. (**a**) TAC (Auto-*m*); (**b**) SDT.

**Figure 4 sensors-21-04153-f004:**
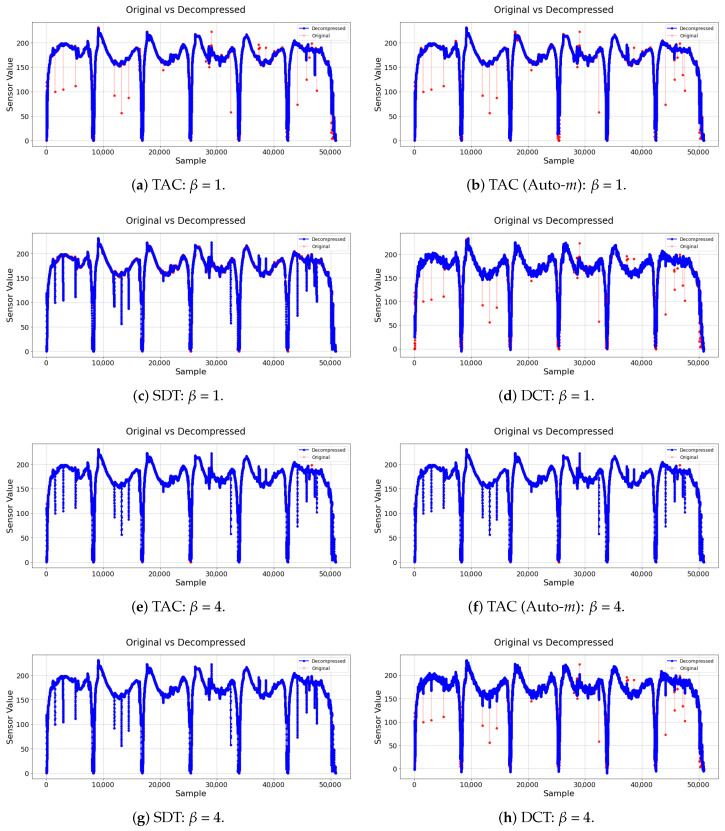
Original and decompressedsignals: Dataset 1.

**Figure 5 sensors-21-04153-f005:**
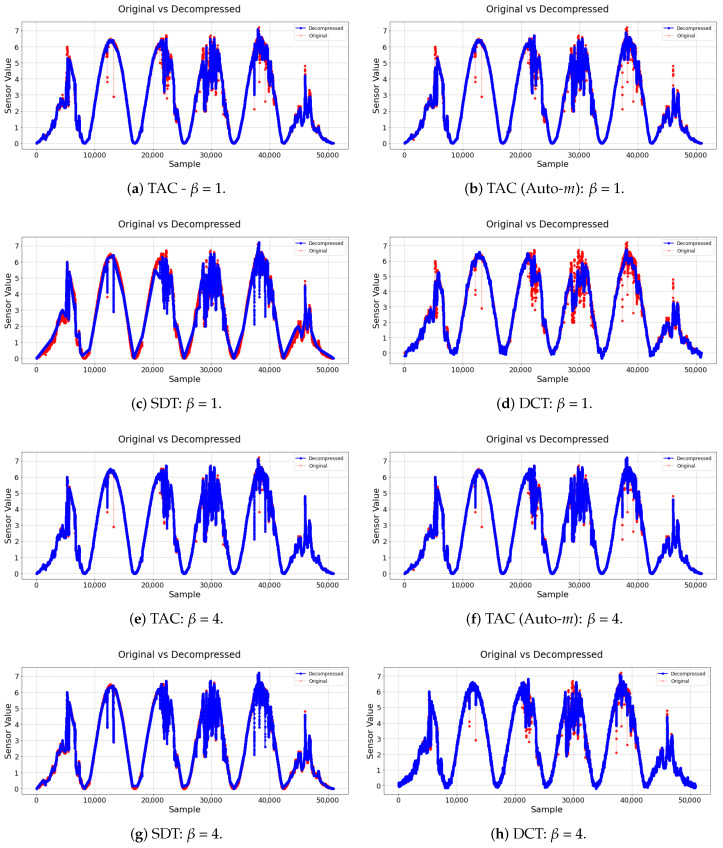
Original and decompressed signals: Dataset 2.

**Table 1 sensors-21-04153-t001:** Summary of related works.

	Features	Online Algorithm	Machine Learning	Lightweight Solution	Data Are Encoded
Works					
Bristol [43]		yes	no	yes	no
Correa et al. [11]		yes *	no	yes	no
Moon et al. [41]		no	no	no	yes
Gibson et al. [46]		yes	no	no	yes
Lounas et al. [10]		no	no	no	yes
Park et al. [47]		no	yes	no	yes
Azar et al. [9]		yes	no	yes	yes
Yu et al. [39]		no	yes	no	yes
Proposed Work		yes	yes	yes	no

* Requires a training phase.

**Table 2 sensors-21-04153-t002:** Basic statistical properties of the dataset considered in the case study.

Dataset	Samples	Min	Max	Mean	Standard Deviation	Skewness
Voltage	50,879	0.0	231.0	169.739	39.198	−2.633
Current	50,879	0.0	7.2	2.627	2.119	0.374

**Table 3 sensors-21-04153-t003:** Optimal hyperparameters: Dataset 1.

Algorithm	Hyperparameters for β=1	Hyperparameters for β=4
TAC	window_limit=35, m=0.2	window_limit=7, m=0.1
TAC (Auto-*m*)	window_limit=32	window_limit=4
SDT	comp_deviation=1.4	comp_deviation=0.4
DCT	delta=0.998	delta=0.999

**Table 4 sensors-21-04153-t004:** Compression results: Dataset 1.

β	Algorithm	CR (%)	NCC	CF${}_{1}$ Score	MSE	MAE	PSNR
1	TAC	97.4	0.9865	0.9802	41.3406	1.4593	31.1085
	TAC (Auto-*m*)	97.49	0.9849	0.9799	46.2367	1.6009	30.6224
	SDT	97.15	0.9807	0.9761	60.2682	2.3412	29.4714
	DCT	98.8	0.9801	0.984	60.6456	4.2092	29.4442
4	TAC	91.5	0.9948	0.9898	16.0263	0.738	35.2239
	TAC (Auto-*m*)	90.31	0.9887	0.9887	16.7838	0.6913	35.0233
	SDT	91.56	0.9896	0.9849	32.0155	1.2412	32.2186
	DCT	96.8	0.9901	0.9888	30.3387	2.9515	32.4523

**Table 5 sensors-21-04153-t005:** Optimal hyperparameters: Dataset 2.

Algorithm	Hyperparameters for β=1	Hyperparameters for β=4
TAC	window_limit=40, m=0.6	window_limit=15, m=0.1
TAC (Auto-*m*)	window_limit=40	window_limit=12
SDT	comp_deviation=0.3	comp_deviation=0.1
DCT	delta=0.992	delta=0.997

**Table 6 sensors-21-04153-t006:** Compression result: Dataset 2.

β	Algorithm	CR (%)	NCC	CF${}_{1}$ Score	MSE	MAE	PSNR
1	TAC	98.33	0.9919	0.9876	0.0723	0.1038	28.5547
	TAC (Auto-*m*)	98.24	0.9916	0.987	0.0753	0.101	28.377
	SDT	98.98	0.9883	0.989	0.1263	0.2671	26.1318
	DCT	99.48	0.9899	0.9923	0.0908	0.1845	27.5638
4	TAC	95.26	0.9972	0.9944	0.0256	0.0542	33.0617
	TAC (Auto-*m*)	95.37	0.9963	0.9937	0.0337	0.0602	31.8695
	SDT	97.2	0.9958	0.9943	0.0392	0.1066	31.2167
	DCT	97.31	0.9962	0.9948	0.0342	0.1117	31.8127

## Data Availability

Not applicable.

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
