# Peer review of "An Evolving TinyML Compression Algorithm for IoT Environments Based on Data Eccentricity"

_sensors, 2021, doi:10.3390/s21124153_

Round 1
Reviewer 1 Report
1. I am not convinced as to why data compression is required for LPWA networks that deploy massive machine-type communication IoT devices. Most of these devices rely on very simple sensors that store/send small amounts of data. Therefore, it is highly unlikely that we would need any form of local data compression. If the authors are referring to specific sensors that rely on huge amounts of data such as cameras for example, then sure I can see the benefits of data compression in these applications. If that is so, the authors need to clearly state that this work is only beneficial to these applications.
2. There are many disadvantages of incorporating data compression in IoT devices and the authors need to address them:
- increases the complexity of the device.
- increases the energy drainage of the battery (for compression computation).
- increases delay prior to transmission.
Regarding the second points, I get that compressing the data would reduce the frame size and therefore “might” reduce the overall energy consumption, however we are still going to lose energy through computation so this needs to be carefully stated. Please note that if compression is implemented in simple sensors, then the trade-off might fail since compression might not lead to significant energy saving when compared to computational energy wastage.
3. I understand that it might outperform legacy methods such as DCT, however, what is the point of using data compression when a significant size reduction is not achieved in IoT applications. A LoRaWAN device can only send up to 255 bytes per payload per packet (or transmission) for example, hence if the proposed compression technique is not capable of achieving significant size reduction, then many packets are still required to be sent (leading to interference, higher duty cycle, and again energy!). Therefore, it is significant that the authors clearly in the abstract/conclusion what the best compression ratio, size of data after compression/size of data before compression, that can be achieved using the TAC methodology proposed in the article to provide more applicability to IoT technologies.
4. The reference list can be significantly enhanced, avoid referencing citations that are yet to be published or are published on arXiv. Below are some references on IoT applications and IoT energy that might be helpful:
- Raza, P. Kulkarni and M. Sooriyabandara, "Low power wide area networks: An overview", IEEE Commun. Surveys Tuts., vol. 19, no. 2, pp. 855-873, 2nd Quart. 2017.
- Al Homssi et al., "A Framework for the Design and Deployment of Large-Scale LPWAN Networks for Smart Cities Applications," in IEEE Internet of Things Magazine, vol. 4, no. 1, pp. 53-59, 2021.
- Zanella et al., "Internet of Things for Smart Cities", IEEE J. Internet of Things, vol. 1, no. 1, pp. 22-32, Feb. 2014.
- Georgiou and U. Raza, "Low power wide area network analysis: Can LoRa scale?", IEEE Wireless Commun. Lett., vol. 6, no. 2, pp. 162-165, Apr. 2017.
5. Section 4 is very well-known and therefore should be drastically reduced (with no equations since they are very well-known) and instead added to related works. DCT is a very well-known method! Instead, the proposed algorithm should come right after Related works.
6. The tables are not a very convenient way of comparing, they are very confusing (also their font size is not equal to the caption). I recommend that the authors think about a better way of representing the outputs of the paper by converting them to visuals instead.
7. The figures are extremely small (the legends, axes ticks, and axes labels are unreadable) in Figures 3-7. Please ensure that they are equal in font size to the captions.
8. Section 6 need to be appropriately renamed. Experiment is usually referred to actual measurements taken in a lab setting while the provided analysis is based on simulations.
9. The article looks like a technical report, this is a journal article and does not require the authors to provide their work in research question format which is typically utilized in theses. The conclusion should be self-explanatory on its own and thus avoid using “Question 1” or “Question 2”. Please use the names of the criteria in the conclusion since it is difficult to go back and forth between the manuscript.
10. There are some grammatical mistakes in the submitted manuscript. I recommend the authors carefully read the paper and revise its language.
Author Response
Dear Editor,
We very much appreciate your helpful comments. All the issues have been addressed as requested. In the following pages, we detail our response to the editor and reviewers’ comments and the changes performed in the new version. We think that the manuscript has been greatly improved by these revisions and we hope that you will now find it suitable for publication in MDPI Sensors.
We are at your disposal for any further clarifications. Please, do not hesitate to contact us.
Sincerely,
Reviewer 1
We apologize for not being able to clearly highlight that the main goal of the paper is to introduce a compressing algorithm that can be used for continuous stream of data in IoT scenarios.
- I am not convinced as to why data compression is required for LPWA networks that deploy massive machine-type communication IoT devices. Most of these devices rely on very simple sensors that store/send small amounts of data. Therefore, it is highly unlikely that we would need any form of local data compression. If the authors are referring to specific sensors that rely on huge amounts of data such as cameras for example, then sure I can see the benefits of data compression in these applications. If that is so, the authors need to clearly state that this work is only beneficial to these applications.
We agree with the reviewer when he states that most IoT nodes are based on devices offering limited resources in terms of computational and storage capabilities. Indeed, one of the goals of the research activity is to find out a compression strategy demanding limited amounts of the previously addressed resources.
However, we think that compression is “likely” to be requested, especially referring to the mMTC communications. As correctly noted by the reviewer himself, mMTC are currently supported by so-called LPWANs. The term is an umbrella for many different technologies. A lot of them operate in unlicensed bands and sub-GHz region of the spectrum is preferred when large area coverage is desired (as for LoRaWAN, SigFox etc). As a consequence, the available bandwidth is limited as well and a reduced number of RF channels is available. Indeed, most of LPWAN solutions offer small data rates, i.e. (very) long time-on-air even for small message payloads.
In turn, it means that in high density scenarios (very likely in IoT applications, e.g., think about Smart “Environments” as for Smart Cities, Smart Transportation etc) scalability issues arise. Such a concern is even worse when simple medium access strategies are applied in order to minimize protocol stack complexity (e.g., despite inefficient, pure ALOHA is a common choice). For this reason, even a small reduction of the payload could provide valuable improvements in the number of nodes able to successfully communicate in the same area, no matter the actual application or the sensor type.
Moreover, this work demonstrates that, if lossy compression is tolerated, very high CR values (higher than 90%) can be obtained despite the reduced computational complexity of the proposed approach. Accordingly, even if evaluating improvements of the overall network availability with a very large number of nodes is out of the paper scope, we are convinced that the discussion about the compression performance clearly highlights possible advantages and justifies the adoption of compression strategies in many of the typical IoT applications.
- There are many disadvantages of incorporating data compression in IoT devices and the authors need to address them:
- increases the complexity of the device.
- increases the energy drainage of the battery (for compression computation).
- increases delay prior to transmission.
Regarding the second points, I get that compressing the data would reduce the frame size and therefore “might” reduce the overall energy consumption, however we are still going to lose energy through computation so this needs to be carefully stated. Please note that if compression is implemented in simple sensors, then the trade-off might fail since compression might not lead to significant energy saving when compared to computational energy wastage.
Referring to the answer to the previous remark, we underline that, from the scalability point of view, the advantage of compressing data is greater when low data rate, simple MAC, communication is used. For this reason, reducing the time-on-air not only provides consumption reduction for the shorter on time of the (power hungry) radio, but also decreases the collision probability, saving the energy for long retransmission and increasing the number of nodes operating in the same area.
Even if the Massive Machine Type Communication (mMTC) are generally delay tolerant, we focused the research on recursive compression algorithms in order to reduce delays. [Note that mMTC has been differentiated from the so-called Ultra Reliable Low Latency Communications (aka URLLC), where the latency is a main concern].
We agree that the device complexity might increase, but we used specialized recursive approaches based on low-complexity arithmetic operations for low resource devices, minimizing also this constraint. For these reasons, in the introduction we clearly highlighted these aspects, to better exemplify possible application areas of compression in IoT scenarios.
- I understand that it might outperform legacy methods such as DCT, however, what is the point of using data compression when a significant size reduction is not achieved in IoT applications. A LoRaWAN device can only send up to 255 bytes per payload per packet (or transmission) for example, hence if the proposed compression technique is not capable of achieving significant size reduction, then many packets are still required to be sent (leading to interference, higher duty cycle, and again energy!). Therefore, it is significant that the authors clearly in the abstract/conclusion what the best compression ratio, size of data after compression/size of data before compression, that can be achieved using the TAC methodology proposed in the article to provide more applicability to IoT technologies.
Significant size reduction is indeed achieved. As can be seen in the results Section, the novel TAC algorithm is capable of achieving compression rates upwards of 90% in the scenarios tested, leaving less than 10% of the samples to be sent, drastically decreasing the need to use the communication interface of the device. However, adhering to the suggestion presented by the reviewer, this information was also highlighted in the work’s abstract, conclusion and results tables. Finally, the optimal compression rate that can be achieved is entirely dependent on the application requirements and tolerance for compression errors, as such, a single universal value can not be presented.
- The reference list can be significantly enhanced, avoid referencing citations that are yet to be published or are published on arXiv. Below are some references on IoT applications and IoT energy that might be helpful:
- Raza, P. Kulkarni and M. Sooriyabandara, "Low power wide area networks: An overview", IEEE Commun. Surveys Tuts., vol. 19, no. 2, pp. 855-873, 2nd Quart. 2017.
- Al Homssi et al., "A Framework for the Design and Deployment of Large-Scale LPWAN Networks for Smart Cities Applications," in IEEE Internet of Things Magazine, vol. 4, no. 1, pp. 53-59, 2021.
- Zanella et al., "Internet of Things for Smart Cities", IEEE J. Internet of Things, vol. 1, no. 1, pp. 22-32, Feb. 2014.
- Georgiou and U. Raza, "Low power wide area network analysis: Can LoRa scale?", IEEE Wireless Commun. Lett., vol. 6, no. 2, pp. 162-165, Apr. 2017.
The references have been updated, removed citations that are yet to be published or are published on arXiv. In addition, we included your suggestions for new references.
- Section 4 is very well-known and therefore should be drastically reduced (with no equations since they are very well-known) and instead added to related works. DCT is a very well-known method! Instead, the proposed algorithm should come right after Related works.
The suggestion of removing SDT and DCT algorithms detailed description from section 4 was also accepted. However, we expanded section 2 (background) with the most highlighted paragraphs about the general workings of the algorithms, so readers from outside the field can still understand. Finally, we renamed section 4 to focus solely on the mathematical description of the metrics used in TAC;
- The tables are not a very convenient way of comparing, they are very confusing (also their font size is not equal to the caption). I recommend that the authors think about a better way of representing the outputs of the paper by converting them to visuals instead.
The cause of confusion is not clear, as the tables present just the results of the metrics that were used for the benchmark evaluation (all of them described in detail in Section 6). We didn’t find an alternative (visual) way of clearly representing all of the values needed in a concise manner and, as there were already a lot of large plots in the text, we opted to keep the tables as means of representing these metrics. However, the table styling and font size were adjusted to hopefully make it more visible and clear.
- The figures are extremely small (the legends, axes ticks, and axes labels are unreadable) in Figures 3-7. Please ensure that they are equal in font size to the captions.
All plots were remade for better legibility. The labels were written larger font sizes and the figures were enlarged as much as the template would allow.
- Section 6 need to be appropriately renamed. Experiment is usually referred to actual measurements taken in a lab setting while the provided analysis is based on simulations.
Renamed section 6 to “Case Study Definition” instead of “Experiment” to be more aligned with the suggestion. Furthermore, Section 6 text was reorganized to be more cohesive (bringing the dataset selection and goal definition to the beginning). We hope these adjustments make the text easier to follow;
- The article looks like a technical report, this is a journal article and does not require the authors to provide their work in research question format which is typically utilized in theses. The conclusion should be self-explanatory on its own and thus avoid using “Question 1” or “Question 2”. Please use the names of the criteria in the conclusion since it is difficult to go back and forth between the manuscript.
As stated above, Section 6 was reorganized and, in doing so, we also rephrased the research questions and incorporated them into the Goal Definition subsection text, removing the “Question” markers. References to “Question 1” and “Question 2” were removed throughout the text. The definition of the benchmark test procedure at the end of the section was also made clearer.
- There are some grammatical mistakes in the submitted manuscript. I recommend the authors carefully read the paper and revise its language.
The English writing was reviewed in all sections to correct mistakes.

Reviewer 2 Report
This paper proposed an evolving TinyML Compresion algorithm TAC based on data eccentricity for IoT. There is little innovation in this paper.
This paper exists some problems.
- In abstract, author first points the difficulty of collecting and storing data in IoT, and proposed the concept of TinyML. Then, author proposed a new approach TAC. But author does not illustrate the reason of proposing TAC and the details of TAC algorithm. The abstract need be further modified.
- The section 4 is the description of comparison algorithms DCT and SDT, and this section can delete.
- Section 5.2 does not point improvements of TAC algorithms.
Author Response
Dear Editor,
We very much appreciate your helpful comments. All the issues have been addressed as requested. In the following pages, we detail our response to the editor and reviewers’ comments and the changes performed in the new version. We think that the manuscript has been greatly improved by these revisions and we hope that you will now find it suitable for publication in MDPI Sensors.
We are at your disposal for any further clarifications. Please, do not hesitate to contact us.
Sincerely,
Reviewer 2
- In abstract, author first points the difficulty of collecting and storing data in IoT, and proposed the concept of TinyML. Then, author proposed a new approach TAC. But author does not illustrate the reason of proposing TAC and the details of TAC algorithm. The abstract need be further modified.
Following the reviewer request, the abstract and the introduction have been modified in order to better clarify the motivations of the work. For instance, we stress that when mMTC (wireless) communications (exploited by a large part of IoT applications) are considered, any message length reduction results into a reduced time-on-air which in turn provides better spectrum usage and permits a higher node density.
- The section 4 is the description of comparison algorithms DCT and SDT, and this section can delete.
The suggestion of removing SDT and DCT algorithms detailed description from section 4 was also accepted. However, we expanded section 2 (background) with a brief set of paragraphs about the general workings of the algorithms, so readers from outside the field can still understand. Finally, we renamed section 4 to focus solely on the mathematical description of the metrics used in TAC;
- Section 5.2 does not point improvements of TAC algorithms.
Section 5 was reorganized and simplified to make the description of the algorithm clearer. A detailed description of the algorithm math and logic is presented, along with its pseudocode. However, as means to make its application context more clear, a new subsection was added at the end of Section 5 containing a sample execution flowchart of the algorithm when used in an IoT device.

Reviewer 3 Report
Authors have highlighted the emerging and core issue, but still there are major issues to be fixed.
Comments
Minor
- Title must be simple, clearer and nicer.
- Spell out each acronym the first time used in the body of the paper. Spell out acronyms in the Abstract by extending it.
- The abstract can be rewritten to be more meaningful. The authors should add more details about their final results in the abstract. Abstract should clarify what is exactly proposed (the technical contribution) and how the proposed approach is validated.
Major
- What is the motivation of the proposed work?
- Introduction needs to explain the main contributions of the work clearer.
- The novelty of this paper is not clear. The difference between present work and previous Works should be highlighted.
- Authors must explain in detail the introduction section.
- Authors must develop the framework/architecture of the proposed methods
- There is need of flowchart and pseudocode of the proposed techniques
- Proposed methods should be compared with the state-of-the-art existing techniques
- Research gaps, objectives of the proposed work should be clearly justified.
- The authors should consider more recent research done in the field of their study for strengthening the Introduction and related work sections < A Novel Adaptive Battery-Aware Algorithm for Data Transmission in IoT-Based Healthcare Applications, Electronics, MDPI, Vol.10, No.4, pp.367, 2021>, <An Energy-Efficient Algorithm for Wearable Electrocardiogram Signal Processing in Ubiquitous Healthcare Applications”, MDPI Sensors Vol.8, No.3, pp.923, 2018 >
- English must be revised throughout the manuscript.
- Limitations and Highlights of the proposed methods must be addressed properly
- Experimental results are not convincing, so authors must give more results to justify their proposal.
Finally, paper needs major improvements
Author Response
Thanks for suggestions, they were well suitable in order to improve the contributions of paper. The authors implemented all suggestions in this version to improve the work in several aspects. In general, the sections were reviewed according to the reviewer comments and the respective modifications are listed as follows (similar remarks were grouped together to simplify the answers):
Review 3
Minor
- Title must be simple, clearer and nicer.
The title was shortened as per the Reviewer suggestion
- Spell out each acronym the first time used in the body of the paper. Spell out acronyms in the Abstract by extending it.
A review regarding acronyms was made throughout the entire text.
- The abstract can be rewritten to be more meaningful. The authors should add more details about their final results in the abstract. Abstract should clarify what is exactly proposed (the technical contribution) and how the proposed approach is validated.
Following the reviewer request, the abstract have been modified in order to better clarify the motivations of the work, obtained results and validation procedure.
Major
- What is the motivation of the proposed work? Introduction needs to explain the main contributions of the work clearer. The novelty of this paper is not clear. The difference between present work and previous Works should be highlighted. Authors must explain in detail the introduction section.
The main motivation for the development of this work comes from the ever-increasing presence of IoT devices in everyday life. With this prevalence, issues of data management and scalability arise. In this context, the work presents a novel compression algorithm developed from the ground up to fit the requirements of these applications. The TAC algorithm is a completely different approach from the other methods cited: it is based on Eccentricity and Typicality, and not on traditional probability mathematics; It is lightweight, as it performs simple recursive arithmetic calculations; It doesn’t need prior training or prior assumptions about the underlying data distributions; and, more importantly, it can be executed online and perform analysis on a point-by-point basis. All of these characteristics, as well as other contributions of the paper, are clearly detailed at the end of the Introduction section.
- Authors must develop the framework/architecture of the proposed methods. There is need of flowchart and pseudocode of the proposed techniques
The detailed description of the algorithm math and logic was already presented in Section 5, along with its pseudocode. However, as means to make its application context and architecture more clear, a new subsection was added at the end of Section 5 containing a sample execution flowchart of the algorithm when used in an IoT device.
- Proposed methods should be compared with the state-of-the-art existing techniques. Research gaps, objectives of the proposed work should be clearly justified.
The main objective of the paper was to describe the novel TAC algorithm in detail and perform a base benchmark test to validate it in contrast with well-documented and established models that can perform the same tasks. It was chosen to perform this test along with the SDT and DCT algorithms in order to make the case study experiments more easily reproducible by other researchers that may want to verify the obtained results. Additionally, as described in Section 3 (related works), most of the state-of-the-art algorithms found in this area suffer from one or more of these problems: Are not lightweight, and thus can only be executed in a computer or cloud data center; are not online and evolving such as the novel TAC algorithm; Need training phases and don’t learn and update continuously (are static); and, finally, most works are not easily reproducible.
- The authors should consider more recent research done in the field of their study for strengthening the Introduction and related work sections < A Novel Adaptive Battery-Aware Algorithm for Data Transmission in IoT-Based Healthcare Applications, Electronics, MDPI, Vol.10, No.4, pp.367, 2021>, <An Energy-Efficient Algorithm for Wearable Electrocardiogram Signal Processing in Ubiquitous Healthcare Applications”, MDPI Sensors Vol.8, No.3, pp.923, 2018 >
The references have been updated and the Reviewer suggestions for new references were included.

Round 2
Reviewer 1 Report
Authors have addressed all my comments.
Author Response
Dear Reviewer,
We very much appreciate your helpful comments. All the issues have been addressed as requested. We think that the manuscript has been greatly improved by these revisions and we hope that you will now find it suitable for publication in MDPI Sensors.
We are at your disposal for any further clarifications. Please, do not hesitate to contact us.
Sincerely,
Reviewer 3 Report
Paper is significantly improved, so I am accepting it
Author Response

(The authors gave the same response as above.)
